# Is Bitcoin’s Carbon Footprint Persistent? Multifractal Evidence and Policy Implications

**DOI:** 10.3390/e24050647

**Published:** 2022-05-05

**Authors:** Bikramaditya Ghosh, Elie Bouri

**Affiliations:** 1Symbiosis Institute of Business Management (SIBM), Symbiosis International (Deemed University) (SIU), Bengaluru 560017, India; 2School of Business, Lebanese American University, Beirut P.O. Box 13-5053, Lebanon

**Keywords:** Bitcoin carbon footprint, Bitcoin mining, energy consumption, FIGARCH, MFDFA, long memory, Hurst exponent, permanent policy

## Abstract

The Bitcoin mining process is energy intensive, which can hamper the much-desired ecological balance. Given that the persistence of high levels of energy consumption of Bitcoin could have permanent policy implications, we examine the presence of long memory in the daily data of the Bitcoin Energy Consumption Index (BECI) (BECI upper bound, BECI lower bound, and BECI average) covering the period 25 February 2017 to 25 January 2022. Employing fractionally integrated GARCH (FIGARCH) and multifractal detrended fluctuation analysis (MFDFA) models to estimate the order of fractional integrating parameter and compute the Hurst exponent, which measures long memory, this study shows that distant series observations are strongly autocorrelated and long memory exists in most cases, although mean-reversion is observed at the first difference of the data series. Such evidence for the profound presence of long memory suggests the suitability of applying permanent policies regarding the use of alternate energy for mining; otherwise, transitory policy would quickly become obsolete. We also suggest the replacement of ‘proof-of-work’ with ‘proof-of-space’ or ‘proof-of-stake’, although with a trade-off (possible security breach) to reduce the carbon footprint, the implementation of direct tax on mining volume, or the mandatory use of carbon credits to restrict the environmental damage.

## 1. Introduction

Bitcoin is a celebrated yet controversial digital currency that continues to attract much attention from users, investors, and regulators across the globe. It is a completely decentralised digital currency without a regulator, with transactions recorded in a publicly distributed ledger called a blockchain [1,2]. New transactions are bucketed into ‘blocks’ and written onto the end of a ‘chain’ of pre-existing blocks representing old transactions, hence the name ‘blockchain’. Despite wide price fluctuations and periods of booms and busts, Bitcoin holds a major volume in the cryptocurrency domain. Notably, new Bitcoin is introduced into circulation via a process called ‘mining’, through which transactions are validated for a blockchain. Successful miners are rewarded newly minted Bitcoin for synchronising Bitcoin transactions after solving a complex hashing puzzle. In the process of such proof-of-work mining, new Bitcoins are issued at intervals of almost 10 min, and finding a single block of Bitcoin involves approximately 10 hash computations. While less energy-intensive mechanisms of mining, such as proof-of-space or proof-of-stake, have recently emerged to secure transactions on blockchain by enabling computer networks to collaborate, their application cannot guarantee security and raises significant technological issues. This is why proof-of-work remains the most popular mechanism of mining.

Bitcoin miners operate specialised mining devices with increasingly advanced hardware, such as application-specific integrated circuits (ASICs). They generally use multiple machines to synchronise Bitcoin transactions and optimise their odds of getting the mining reward, which requires repeatedly running and cooling multiple mining machines. Notably, enormous energy resources are wasted in the Bitcoin mining process. In fact, the mining process consumes huge amounts of electricity [3], and the resulting electricity consumption has been measured at 110.53 TWh per year, exceeding the energy consumption of the Netherlands, inducing a carbon footprint of 36.95 megatons of CO_2_ per year, comparable to that of New Zealand. Rising Bitcoin prices make mining very lucrative and attractive, which leads to more electricity consumption [1,4] and greater carbon footprints.

Previous studies show that electricity consumption has a direct and positive relationship with CO_2_ emissions [5], and a study of Chinese and Russian electricity markets finds that Bitcoin price volatility is positively correlated with the utility market pricing volatility [6]. The public transaction record (blockchain) is also very energy intensive. However, proof-of-work consumes significantly more energy than proof-of-stake. The number of miners may decline over time or move to more energy-efficient machines [7]. 

In this paper, we contribute to the above debate on the carbon footprint of Bitcoin mining by examining the long memory process in Bitcoin electricity consumption that reflects the CO_2_ emissions of the mining process. Specifically, we use daily data on the Bitcoin Energy Consumption Index (BECI) over the period 25 February 2017 to 25 January 2022 and apply fractionally integrated GARCH (FIGARCH) models and multifractal detrended fluctuation analysis (MFDFA).

Accordingly, we extend the above literature, which remains silent on whether Bitcoin’s carbon footprint, measured by energy consumption, exhibits a long memory process. Interestingly, the true long memory process has many facets, which make its application to Bitcoin electricity consumption very informative. Its implications matter to the choice of the most suitable policies that should be applied to address the carbon footprint of Bitcoin mining. For linking stationary long memory and the types of policy (transitory versus permanent), Belbute & Pereira [6,8] argue that if emissions are stationary, then transitory policies (i.e., promotion of energy efficiency, switching from fossil fuel to green energy etc.) will have only transitory effects and fade away in the long-term. Conversely, if emissions are non-stationary, then transitory policies will have a lasting permanent effect [6,8]. Concerns over the energy consumption of Bitcoin mining are indicated by McCook [9], who includes mining-rig procurement and cooling calculations, and argues that Bitcoin is less harmful to the environment than gold mining. Bitcoin’s carbon footprint is comparable to that of Ireland [10], and Mora et al. [11] confirm that the estimated CO_2_ emissions from Bitcoin could make the globe warmer by 2 °C. Howson [12] expresses concern about the carbon footprint of Bitcoin, while Krause and Tolaymat [10] show that the mining of 4 cryptocurrencies (Bitcoin, Ethereum, Litecoin, and Monero) generated 3–15 million tonnes of CO_2_ emissions over the period 1 January 2016 to 30 June 2018. Sedlmeir [13] points to the huge energy consumption of blockchains, especially on the basis of the number of transactions they operate.

In fact, a strong statistical dependence of a mean-reverting time series indicates long memory, long-range dependence or simply persistence [14,15,16]. Generally, the dependence becomes weaker with time but not in the presence of long memory. Non-stationary time series also show evidence of persistence, sometimes even more strongly than stationary series. Thus, mean reversion holds the key to true long memory. Past and present values are connected by a fractionally integrating parameter, d, which must be empirically calibrated. A partially long memory exists when d ≠ 0, since a significant mean reversion happens at first difference. For such cases, permanent policy changes are recommended to address the carbon footprint of Bitcoin mining. However, for pure long memory (d = 0), the effects generated by transitory policy shocks persist for a long time, and thereby the type of long memory indicates the preferred type of policy that should be adopted by regulators and policymakers to address the carbon footprint of Bitcoin mining.

Following this introduction, Section 2 describes the data and methodology. Section 3 presents and discusses the results. Section 4 concludes and offers policy implications.

## 2. Data and Methodology

### 2.1. Data

We used daily data from the Bitcoin Energy Consumption Index (BECI) over the period 25 February 2017 to 25 January 2022, according to the data availability from Digiconomist. BECI data have been recently used in academia [12,17]. They consist of daily data covering three series, BECI upper bound (BECI UB), BECI lower bound (BECI LB), and BECI average. Accordingly, in this work, 1825 daily observations were used for each of the three series. The plots of the three series are shown in Figure 1, in which an increase in the three series is observed from around the second quarter of 2021, which coincides with the spike in the price of Bitcoin.

Considering both BECI upper bound (BECI UB) and BECI lower bound (BECI LB) for electricity consumptions, we argue the following. The BECI UB is defined as the break-even point of mining revenues and electricity costs; therefore, it is more sensitive to the economic parameters. In contrast, the BECI LB is a state where all miners use the most efficient hardware, which makes it more stable and reliable for our current study [13,14] examining the long memory traits of the energy consumption series. Therefore, our analysis emphasises the lower bound results.

Our analysis involved the application of 17 windows to the three series, with the length of each sliding window being 200 days. Given that each series consists of 3400 (17*200) daily observations, a total of 51(17*3) windows (with 10,200 total observations, 3400*3) were considered in our calculation for the sample period 25 February 2017 to 25 January 2022. The choice of sliding window-based estimation procedure is backed by the academic literature [18], which points to the suitability of the application of an increasing window size in a dynamic model for estimating long memory. A sliding window approach to modelling is suitable and an increasing estimation window leads to an increase in the estimation accuracy when calibrating the long memory [19].

Each series observation in the BECI indices is expressed in terawatt-hours (TWh), the standard unit of electricity consumption. The three BECI series are significantly mean-reverting at first difference. Therefore, all the calculations and analyses were conducted at first difference (Δ). Long-range dependence or long memory was characterised by a slow, power law decay of the autocorrelation function (ACF).

### 2.2. Methodology

Methodologically, we used a pluralistic approach combining two different methods to calibrate ‘d’ and ‘H’. The FIGARCH is an extension of the famed GARCH family as described by Baillie (1996), which is consistent with EMH [20]. The MFDFA is an extension of detrended fluctuation analysis (DFA) as described by Kantelhardt (2008), which is consistent with FMH [21,22].

Firstly, we employed a FIGARCH model to uncover evidence of long memory in BECI indices. FIGARCH considers conditional heteroscedasticity and is comparable to ARCH, but allows for long memory in the conditional variance. It is preferred over autoregressive fractionally integrated moving average (ARFIMA) models because it can detect mean-reverting long memory. Usually, financial time series have d = 1 (fractional integrating parameter), which is consistent for log closing prices of various tradeable securities. Furthermore, it is perfectly in harmony with the efficient market hypothesis (EMH), which concludes that closing levels are martingales, and log returns are martingale differences (usually first difference). Martingales are sequences of random variables with the future expectation equaling the present value. Squared returns typically carry a fractional value of *d*.

Consider a time series, such as the first level difference of each of the three BECI indices:(1)$${\Delta \mathrm{BECI}}_{t}=\mu +\varepsilon_{t}\;\;\;\mathrm{with}\;\varepsilon_{t}=\nu_{t}\sigma_{t}^{2}$$
where $\nu_{t}$ is a serially uncorrelated process with zero-mean and unit variance; $\sigma_{t}$ is a time-varying measurable function with respect to the information set available at time t−1 (ψ− (t−1)); and $\sigma_{t}^{2}$ is the time dependent conditional variance of ${\Delta \mathrm{BECI}}_{t}$. The FIGARCH model of Baillie and his co-researchers [20] is given by:(2)$$\left(1-\beta_{1}L\right)\sigma_{t}^{2}=\omega_{0}+\left[1-\beta_{1}L-\alpha_{1}L{\left(1-L\right)}^{d}\right]\varepsilon_{t}^{2}$$
where, 0 ≤ *d* ≤ 1 is the fractional differential (long memory) parameter; *L* is the lag operator; β(L) is a finite order lag polynomial with the roots assumed to be situated outside the unit circle; and $\propto_{k\;}$ represents the autoregressive coefficient of an ARFIMA (1, *d*, 0) model. Unlike ordinary ARCH and GARCH, the FIGARCH model does not reach a constant level quickly. It is reduced to a standard GARCH when d = 0 and to an integrated GARCH (IGARCH) when *d* = 1.

Secondly, we used multifractal detrended fluctuation analysis (MF-DFA) to find h(q) value, where ‘h’ is the Hurst Exponent and ‘q’ is the order [23,24]. To this end, we relied on Espen Ihlen’s algorithm in MATLAB 13 [25]. It involves a five-step process, as follows:
Determining the profile
(3)$$Y\left(i\right)=\sum_{k=1}^{i}\left[x_{k}-\langle x\rangle \right]$$
where, $x_{k}$ is the series, and mean subtraction occurs. Further, i=1,2,….,NDividing the profile: To divide the profile Y(i) N numbers of non-overlapping series of the same length ‘s’. Since *N* may not be a multiple of the time scale ‘s’, $2N_{s}$ was considered.Calculation of the local trend: Local trend finding for each $2N_{s}$ segments are carried out by a least-square fit procedure & finding the variance in this process.
(4)$$F^{2}(S,v)\equiv \;1/s\;\sum_{i=1}^{s}{\left\{Y\left[\left(v-1\right)s+i\right]-y_{v\;}\left(i\right)\right\}}^{2}$$
where $y_{v\;}\left(i\right)$ is the curve fitting polynomial is segment v.
Averaging across all segments to find *q*th order fluctuation function:(5)$$F_{q\;}(S)\;\equiv \;{\left\{\frac{1}{2N_{s}}\;\sum_{v=1}^{2N_{s}}{\left[F^{2}\left(s,v\right)\right]}^{q/2}\right\}}^{1/q}$$
where *q* can be any real number, but not zero. It is interesting to note that *q* = 2 coincides with the standard DFA process. Research suggests that extremely large *q* values (−10 or +10) increase the error in the multifractal spectrum tails [26]; therefore, *q* = 5 was used to calibrate such series, which is recommended by another research work [27].Determination of the scaling property of the fluctuation function:(6)$$\;\;\;\;\;\;\;\;\;\;\;\;\;\;\;\;\;\;\;F_{q\;}(s)\sim \;s^{H\left(q\right)}$$
where *H*(*q*) represents the generalised Hurst exponent of the underlying series.

To understand this process intuitively, we referred to Mandelbrot’s research, according to which, scaling exponents are unique in nature and depend upon time. Hence, monofractal is not a full proof. It depicts an incorrect narrative. Stochastic time series, such as Bitcoin energy consumption (BECI), have multiple dimensions that add further complexity. For this reason, multifractal is preferred over monofractal. Asset returns tend to deviate from the normal distribution. Moreover, they tend to obey Lévy stable condition. In other words, α ranges from 0 to 2, where α = 2 satisfies the condition for Gaussian distribution. Thus, [28] reformulate the Rescaled Range Analysis (R/S) approach proposed by Hurst in 1951. The Hurst exponent expresses H = 1/α; when α = 2, it becomes stochastic; i.e., it follows a Brownian motion. The legacy of fractals was investigated by a group of researchers [29] who constructed the mathematical formulae to measure the impact of multifractality in a noisy time series. Time series with consistent noises can be transformed into ‘random walk’ series by subtracting the mean value [25]. Ihlen (2012) integrates it further. According to his process, calculation for the root mean square variation (RMS) is crucial. RMS values are calculated for the localised areas with clear patterns or trends. Finally, all these RMS values are summarised. These RMS samples usually exhibit ‘power law’ characteristics. In technical terms, this process is known as detrended fluctuation analysis (DFA). The exponent for this relation is the Hurst exponent [29]. Kantelhardt (2002) formulated MFDFA formally for calibration. Ihlen (2012) extended this calculation to the qth-order, suggesting the multifractal detrended fluctuation analysis (MFDFA). Multifractal power law has more than one exponent. Further, Power law relationship and persistent pattern in most cases are two important facets of time series [30,31], such as BECI. The Hurst exponent and fractal dimensions do change from monofractal to multifractal, with the latter being more reliable [32].

Since our data points were 200 for each sliding panel, we altered the segments and scale in the MATLAB code proposed by Ihlen (2012). We took segment = 200 and scale = 4. In the first loop, samples 0–200 (Window 1) were taken. In the second loop, samples 100–300 (Window 2) were considered; the third loop considered 200–400 (Window 3), etc. A polynomial trend fit in each loop was conducted. Quadratic and cubic polynomials were used in this code. We obtained values for the fifth order (*q* = 5) Hurst exponent, and considered it for interpretations as suggested by Kantelhardt. It has recently been proved that a window size of 288, with four sub-windows having 72 observations each, works well through MFDFA [33]; therefore, our window size qualified for a robust calibration.

## 3. Results

Looking at Table 1, the first differences in the three BECI indices are non-normally distributed, as evidenced by the Jarque–Bera statistics, with low kurtosis points to low volatility and high persistence, which can be confirmed by both the fractional integrating parameter and the Hurst exponent (H). The results of the augmented Dickey–Fuller (ADF) test show that a significant level of stationarity is achieved at first difference.

Table 2 gives the ranges of d and H and their interpretation, indicating the difference between intermediate memory tending towards short and long memory.

Table 3 shows that all the estimated d parameters are below the 0.5 level. This shows that there are no cases of non-stationarity (*d* = 1). At the same time, there is no purely stationary case (*d* = 0). All cases are mean-reverting (*d* < 1), but they have different degrees of decaying autocorrelations.

### 3.1. Results from the MFDFA

The results from the MFDFA investigation revealed evidence of mean-reverting long memory process across BECI UB, LB, and average (see Table 4). Further, no pure short memory was detected (*d* = 0). The effect of long memory was found to be consistent across most sliding windows, barring one (window 10). The persistent pattern was observed for 17/17 (100%) cases in BECI UB and 16/17 (94%) cases in BECI LB. Most importantly, all these cases recorded a Hurst exponent significantly larger than 0.5, indicating a consistently higher intensity of long memory. Extremely strong long memory (H > 0.85) was found in more than 90% cases through this method. Multifractal or singularity spectrum was used to describe the fractal dimension (d) having the same HÖlder exponent. Here, all of the empirical values of ‘d’ ranged between (−0.07 to +0.50). Both Figure 2 and Figure 3 (chosen randomly out of 51 Windows in consideration), exhibit similar multifractal spectrums, peak around Hq = 1.1 to 1.3. This is more consistent with FMH rather than EMH, which would suggest a peak around Hq = 0.5. The spectrum in each case ranges from 0.6 to 0.8 on the lower side and from 1.6 to 1.7 on the higher side. Small fluctuations are clearly persistent from these two figures. The lower limit of the spectrums in most cases (barring one) ended around Hq > 0.5, indicating the presence of long memory throughout. Another interesting observation is related to the width of the multifractal spectrum. In all 51 windows the width (multifractal) was around 1, exhibiting clear asymmetry (more for BECI LB). This indicates that the degree of complexity in BECI is quite high. This finding remains consistent with existing research [34]. However, future research may further investigate the differences between the left side (small fluctuations from individual cascades reaching white noise faster) and the right side (contracts far sooner to monofractal) [35].

### 3.2. Results from the FIGARCH

Both BECI UB d and BECI LB d indicated a completely stationary process with long memory (see Table 3). The BECI average likewise exhibits a long memory process. However, no cases of short memory (*d* = 0) were detected. Logically, the effect of long memory changes with the sliding window. The results are consistent in terms of the Hurst exponent (except for one instance, where BECI LB showed anti-persistence). BECI UB displayed a persistent pattern in 17/17 cases, whereas BECI LB showed persistent values in 16/17 cases. The BECI average showed a clear persistent pattern in 17/17 cases with the H value being higher than 0.5. A notable 88% of the BECI UB indicated extremely long memory (H > 0.85), whereas BECI average stood second, with 82%, followed by BECI LB, with 76%. Given that hardware costs are substantial and electricity prices are not constant (globally), upper-bound-based calibration is sensitive to various cost matrices [13]. Accordingly, the BECI UB cannot be taken too seriously when making policy decisions. The BECI LB can be considered a better proxy of electricity consumption, and thus suitable for measuring the carbon footprint of the Bitcoin market.

### 3.3. Overall Results Analysis

It can be concluded that Bitcoin energy consumption (represented by the BECI LB) exhibits strong traces of long memory, with 90% of the feature (i.e., Hurst exponent) scoring over 0.85 with the same argument. Both the fractional integration parameter, d, and the measurement of long memory, H, provide enough evidence of true long memory with mean-reverting traits in the Bitcoin carbon footprint (as represented by BECI LB). Hence, Bitcoin’s electricity consumption-led carbon footprint has an overall persistent pattern, with varying degrees across different window sizes. These findings complement the existing literature [36,37,38,39], and provide evidence for the suitability of applying permanent policy implications to address the carbon footprint of Bitcoin mining. The MFDFA findings were consistent with FMH and were found to be more realistic; they uncover a higher degree of long memory over FIGARCH (the traditional model in accordance with EMH).

## 4. Conclusions and Policy Implications

In this paper, we examine the long memory process of the three series of the Bitcoin Energy Consumption Index (BECI) from a pluralistic viewpoint (using FIGARCH and MFDFA) to make inferences regarding the carbon footprint of Bitcoin mining and the possible long-lasting impact of various energy efficiency programs.

Using daily data from 25 February 2017 to 25 January 2022 and FIGARCH model, the results show evidence for the presence of long memory in most series, although with varying degrees. Since the three BECI indices are stationary at first difference (d < 1 across all windows), transitory policy changes to reduce the carbon footprint cannot be sustained in the long-term, given that intuitively random policy shocks do not sustain for a longer period in any mean-reverting time series (stationary). Conversely, a permanent policy implementation would have a much longer-lasting effect. Accordingly, permanent policy implementation becomes an evident course of action. Furthermore, finding alternative energy sources, or applying carbon-footprint reduction policies to Bitcoin mining, becomes imperative.

Some policy alternatives emerge in the light of these findings. Specific types of low-energy-consuming hardware for mining Bitcoin using proof-of-stake should replace the energy-consuming proof-of-work scheme by way of a declared policy. Accordingly, selective miners with proof of low energy usage could be allowed to conduct the mining. Proof-of-space, requiring a defined amount of memory, can be put to use by policymakers, which might be far less energy consuming than even proof-of-stake. However, security issues remain for both proof-of-space and proof-of-stake, indicating a clear trade-off. Although some researchers argue that private or consortium blockchains could be used instead of a public blockchain, it is worth noting that if the major participants reach such an agreement, private or consortium blockchain transactions would become perfectly editable (if they use proof-of-stake or proof-of-space). This presents a major drawback from a security perspective (especially for financial and other sensitive cases) but would make the blockchain system more energy efficient. Permanent environmental policies are crucial in this regard since they would be more effective due to the persistent pattern of the carbon footprint. Accordingly, green energy could be made mandatory for Bitcoin mining. Alternatively, direct tax could be levied on mining volume to further restrict or contain it.

As Bitcoin is contributing to the production of an environmental crisis, future research could consider whether the entire cryptocurrency universe, including major cryptocurrencies other than Bitcoin, is also responsible for high energy consumption and a large carbon footprint. Carbon credits could be made mandatory (they are voluntary as of now) for cryptocurrency miners, forcing responsible mining. Intuitively, efficient and relatively mature markets have seasoned underlying energy efficiency, which propels them to stay afloat in the long term. Currently, most cryptocurrencies suffer from excess energy consumption. Potentially, an alternate technology other than blockchain may be needed to take Bitcoin to a truly mature stage, given that blockchain seems to suffer from extreme consumption of energy, inability to correct its protocol midway (thus abandoning the entire chain), and relatively low performance (5 transactions/second compared to 1700 transactions/second by Visa) [40]. Moreover, the movement of Bitcoin too is not consistent with the EMH [41], making it even more difficult to predict. Hopefully, the tight competition for Bitcoin mining rewards will spur a technological evolution and thereby a genuine solution to the energy-intensive mining process of Bitcoin.

## 5. Limitations & Future Scope of Study

We found some limitations and future scope of study during our investigation. First, there is a need to modify Ihlen (2012) code, in order for Fq to remain well within the segment. Second, conclusive evidence behind the selection of qth order (2 or 5) must be sought. Third, the asymmetry coefficient could be calibrated separately for both sides, as they have different meanings. 

## Figures and Tables

**Figure 1 entropy-24-00647-f001:**
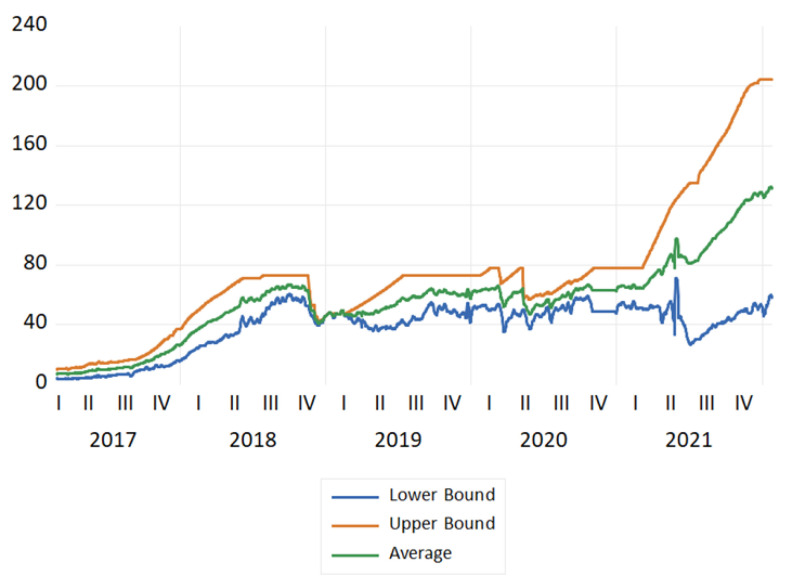
Plots of BECI UB, LB, and average during the study period (25 February 2017 to 25 January 2022).

**Figure 2 entropy-24-00647-f002:**
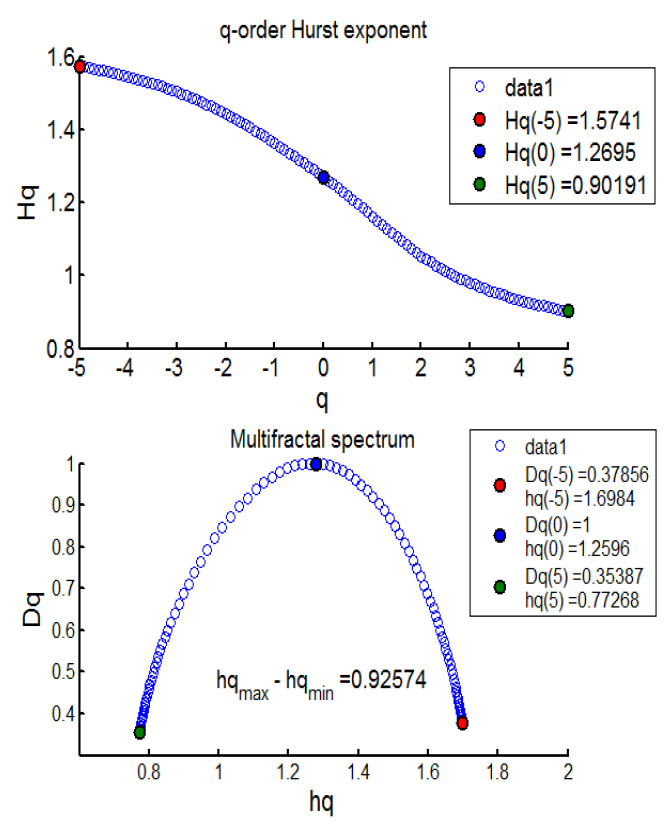
BECI UB window 1 exhibiting the Multifractal Spectrum.

**Figure 3 entropy-24-00647-f003:**
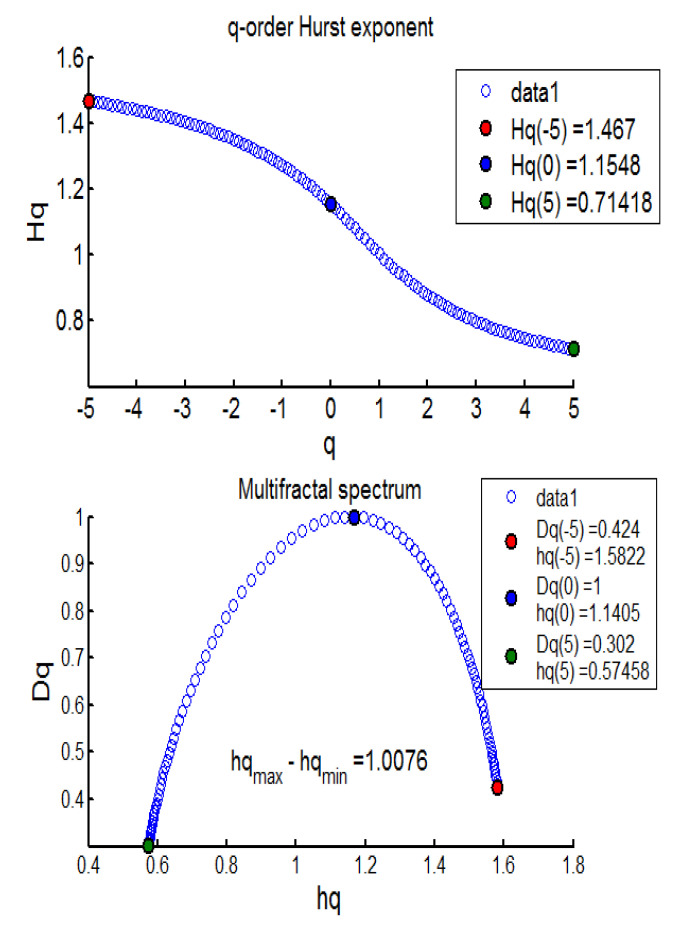
BECI LB window 14 exhibiting the Multifractal Spectrum.

**Table 1 entropy-24-00647-t001:** Summary statistics of the first difference of BECI indices.

	Mean	Max.	Min.	Std. Dev.	Kurtosis	Jarque-Bera	ADF Test
BECI-LB	0.0261	0.818	−0.801	0.200	6.871	124.51	−16.451 *
BECI-UB	0.0349	0.5128	−0.587	0.139	5.892	72.61	−13.017 *
BECI Average	0.0305	0.5293	−0.463	0.131	6.098	86.18	−12.071 *

Notes: The sample period is 25th February 2017 to 25th January 2022. BECI upper bound (BECI UB), BECI lower bound (BECI LB). * Indicates statistical significance at the 1% level.

**Table 2 entropy-24-00647-t002:** Ranges of d and H and their interpretations.

Ranges of ‘d’	Ranges of ‘H’	Interpretation
−0.5 < d < 0	0 < H < 0.5	Intermediate memory tending towards short memory
0 < d < 0.5	0.5 < H < 1	Long memory, autoregression decays

Notes: d is the fractional differential (long memory) parameter. H stands for Hurst exponent, which measures the extent of long memory in time series.

**Table 3 entropy-24-00647-t003:** d and H values for BECI UB, LB and Average- FIGARCH method.

Window Number	Sliding Observations	BECI UB d	BECI UB H	BECI LB d	BECI LB H	BECI Average d	BECI Average H
1	0–200	0.36	0.86	0.50	1.00	0.43	0.93
2	100–300	0.45	0.95	0.41	0.91	0.43	0.93
3	200–400	0.35	0.85	0.30	0.80	0.33	0.83
4	300–500	0.48	0.98	0.41	0.91	0.45	0.95
5	400–600	0.44	0.94	0.41	0.91	0.42	0.92
6	500–700	0.44	0.94	0.31	0.81	0.37	0.87
7	600–800	0.42	0.92	0.29	0.79	0.36	0.86
8	700–900	0.45	0.95	−0.05	0.45	0.20	0.70
9	800–1000	0.41	0.91	0.13	0.63	0.27	0.77
10	900–1100	0.46	0.96	0.43	0.93	0.44	0.94
11	1000–1200	0.29	0.79	0.48	0.98	0.39	0.89
12	1100–1300	0.50	1.00	0.38	0.88	0.44	0.94
13	1200–1400	0.33	0.83	0.47	0.97	0.40	0.90
14	1300–1500	0.43	0.93	0.40	0.90	0.42	0.92
15	1400–1600	0.41	0.91	0.48	0.98	0.45	0.95
16	1500–1700	0.43	0.93	0.43	0.93	0.43	0.93
17	1600–1800	0.43	0.93	0.37	0.87	0.40	0.90

Note: This table shows evidence of more observations having long memory using FIGARCH, but of various degrees. d is the fractional differential (long memory) parameter. H stands for Hurst exponent, which measures the extent of long memory in time series. BECI (Bitcoin Energy Consumption Index), BECI UB (BECI upper bound), BECI LB (BECI lower bound), and BECI Average. The sample period is 25 February 2017 to 25 January 2022.

**Table 4 entropy-24-00647-t004:** d and H values for BECI UB, LB and Average- MFDFA method.

Window Number	Sliding Observations	BECI UB d	BECI UB H	BECI LB d	BECI LB H	BECI Average d	BECI Average H
1	0–200	0.40	0.90	0.27	0.77	0.34	0.84
2	100–300	0.40	0.90	0.35	0.85	0.38	0.88
3	200–400	0.48	0.98	0.45	0.95	0.47	0.97
4	300–500	0.45	0.95	0.34	0.84	0.40	0.90
5	400–600	0.35	0.85	0.39	0.89	0.37	0.87
6	500–700	0.34	0.84	0.44	0.94	0.39	0.89
7	600–800	0.32	0.82	0.09	0.59	0.21	0.71
8	700–900	0.49	0.99	0.45	0.95	0.47	0.97
9	800–1000	0.48	0.98	0.41	0.91	0.45	0.95
10	900–1100	0.35	0.85	−0.07	0.43	0.14	0.64
11	1000–1200	0.40	0.90	0.12	0.62	0.26	0.76
12	1100–1300	0.21	0.71	0.09	0.59	0.15	0.65
13	1200–1400	0.47	0.97	0.27	0.77	0.37	0.87
14	1300–1500	0.32	0.82	0.2	0.7	0.26	0.76
15	1400–1600	0.50	1.00	0.02	0.52	0.26	0.76
16	1500–1700	0.39	0.89	0.26	0.76	0.33	0.83
17	1600–1800	0.47	0.97	0.27	0.77	0.37	0.87

Note: This table shows evidence of more observations having long memory using MFDFA (*q* = 5th order), but of various degrees. d is the fractional differential (long memory) parameter. H stands for Hurst exponent, which measures the extent of long memory in time series. BECI (Bitcoin Energy Consumption Index), BECI UB (BECI upper bound), BECI LB (BECI lower bound), and BECI Average. The sample period is 25 February 2017 to 25 January 2022.

## Data Availability

Data is available on request.

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
