# Peer review of "Is Bitcoin’s Carbon Footprint Persistent? Multifractal Evidence and Policy Implications"

_entropy, 2022, doi:10.3390/e24050647_

Round 1

Reviewer 1 Report

When I went through the paper I thought that it was well written and clear.
In my opinion the topic is of high interest (very limited study on this) and the I believed the methodological approach was interesting to solve the problem with specific reference to the estimation of carbon footprint.
In my opinion the paper is good enough to be published in its current Form. 
I believe other reviewers may have different opinions.
I have replied according to my background which is focused on Footprint modelling and assessment.

Author Response

Dear Editor/Reviewer,

Thank you so much for your remarks. It really motivated us.

Sincerely

Authors

Reviewer 2 Report

The paper addresses analysis of the daily data of the BECI index, which shows the energy consumption of the Bitcoin network. This is an interesting issue; however, the quality of the paper has to be significantly improved before a final decision on acceptance can be made. I listed points to improve below. 

1) Line 111. The authors consider the daily data of BECI from Feb 2017 to Jan 2022. Why do these time series have a length of 3400 observations each? Why not ~1800 (5year*360)?

2) Line 175. Here the generalized Hurst exponent H(q) is mentioned and formally introduced in equation (6) but denoted by h(q) there. Please use a consistent set of symbols. 

3) Line 221. MFDFA algorithm was introduced by J. Kantelhard in 2002 and not by E. Ihlen (cf. doi.org/10.1016/S0378-4371(02)01383-3). Please correct the corresponding information. 

4) Line 227. What are the "segments" (number of scales?) and "scale"?
Can you indicate the quantities in the MFDFA procedure?

5) Line 273. The authors can assess the degree of asymmetry more precisely. The asymmetry coefficient for MF spectrum has been proposed here doi.org/10.1103/PhysRevE.91.030902 

6) Figure 2 and 3. I see that scaling function Fq is estimated on scales much larger than 200 (even 1024). However, the length of the sliding window is only 200 points. Thus, it is impossible to calculate Fq for a scale larger than 200. Moreover, the scaling quality is extremely poor in the presented examples, which raises questions about the fractality of the considered data (can you show the scaling for q=2?). This point has to clarify with particular thoroughness.

In summary, the paper can be considered for publication after a major revision, as indicated above.  

Author Response

Dear Editor/ Reviewer,

We have attached the review report below. We thank you profusely for making our work better.

Sincerely

Authors

This manuscript is a resubmission of an earlier submission. The following is a list of the peer review reports and author responses from that submission.